# Is the *C*-Terminal Domain an Effective and Selective Target for the Design of Hsp90 Inhibitors against *Candida* Yeast?

**DOI:** 10.3390/microorganisms11122837

**Published:** 2023-11-22

**Authors:** Célia Rouges, Mohammad Asad, Adèle D. Laurent, Pascal Marchand, Patrice Le Pape

**Affiliations:** 1Nantes Université, CHU Nantes, Cibles et Médicaments des Infections et de l’Immunité, IICiMed, UR 1155, F-44000 Nantes, France; celia.rouges@aphp.fr (C.R.); pascal.marchand@univ-nantes.fr (P.M.); 2Nantes Université, CNRS, CEISAM, UMR 6230, F-44000 Nantes, France

**Keywords:** *Candida* sp., Hsp90, antifungal targets

## Abstract

Improving the armamentarium to treat invasive candidiasis has become necessary to overcome drug resistance and the lack of alternative therapy. In the pathogenic fungus *Candida albicans*, the 90-kDa Heat-Shock Protein (Hsp90) has been described as a major regulator of virulence and resistance, offering a promising target. Some human Hsp90 inhibitors have shown activity against *Candida* spp. in vitro, but host toxicity has limited their use as antifungal drugs. The conservation of Hsp90 across all species leads to selectivity issues. To assess the potential of Hsp90 as a druggable antifungal target, the activity of nine structurally unrelated Hsp90 inhibitors with different binding domains was evaluated against a panel of *Candida* clinical isolates. The Hsp90 sequences from human and yeast species were aligned. Despite the degree of similarity between human and yeast *N*-terminal domain residues, the in vitro activities measured for the inhibitors interacting with this domain were not reproducible against all *Candida* species. Moreover, the inhibitors binding to the *C*-terminal domain (CTD) did not show any antifungal activity, with the exception of one of them. Given the greater sequence divergence in this domain, the identification of selective CTD inhibitors of fungal Hsp90 could be a promising strategy for the development of innovative antifungal drugs.

## 1. Introduction

Over the past decade, the incidence of systemic candidiasis has increased, making the yeast *Candida albicans* the fourth most common pathogen causing hospital-acquired infections [1]. With exposure to antifungal drugs, the epidemiology of *Candida* species has evolved [2], and the emergence of resistance has become a growing health concern [3]. In addition to widespread fluconazole resistance, echinocandin resistance has also emerged [4,5]. In October 2022, the WHO issued its first fungal priority pathogens list, which included azole-resistant *Candida* spp., among others [6]. Furthermore, host toxicity, drug–drug interactions, and the lack of established and novel therapeutic targets limit current treatments and alternative strategies. To overcome these limitations and improve the therapeutic armamentarium, original targets and relevant compounds need to be identified to develop new antifungal treatments.

The Hsp90 is a homodimeric ATPase of the GHKL (Gyrase, Hsp90, Histidine Kinase, MutL) superfamily, and each monomer consists of three highly conserved domains [7]. The *N*-terminal domain (NTD) contains the ATP-binding site with the lid segment. NTD has negligible catalytic activity alone and is linked to the middle domain by a flexible charged linker. The middle domain (MD) modulates the ATPase activity and interacts with client proteins [8]. The *C*-terminal domain (CTD) is necessary for Hsp90 dimerization. CTD interacts with client proteins and ends with an MEEVD pentapeptide binding with the TPR-domains (TetratricoPeptide-containing Repeats) of co-chaperones. A secondary nucleotide-binding site has been identified in the CTD of Hsp90 [9]. The release of a mature client protein requires a precise cycle of Hsp90 conformational changes driven by intersegment communication, interaction with co-chaperones, interaction with native proteins, and ATP binding [10]. Disruption of this machinery prevents the proper folding and maturation of clients, leading to their destruction by the proteasome. However, many Hsp90 clients (including transcription factors, steroid hormone receptors, and kinases) are involved in critical steps for cell survival, such as cell cycle control and intracellular signaling pathways.

In humans, Hsp90 client proteins are known to play a role in oncogenic processes, making the chaperone a prime target for the development of antitumor drugs [11]. The high abundance of Hsp90 in eukaryotic cells and its phylogenetic interspecies conservation have resulted in growing interest in this target for a number of other diseases, including neurodegenerative diseases [12] and infectious diseases [13,14].

Over 15 years, several human Hsp90 inhibitors have been developed, and 20 of them have already been tested in clinical trials for hematology and cancer therapy. Various generations and classes of Hsp90 inhibitors can be distinguished (Appendix A, https://clinicaltrials.gov consulted on 10 May 2021). The first generation of drugs in trials is the natural benzoquinone ansamycin geldanamycin (GA) and its derivatives, such as 17-AAG, 17-DMAG, IPI-493, and IPI-504. A second generation of non-geldanamycin Hsp90-NTD inhibitors was designed [11]. The most represented class in trials was generated based on the activity of the natural macrocyclic lactone antibiotic radicicol (RA) [15,16]. Pharmacomodulations based on the resorcinol ring permitted the development of a series of clinical candidates, including NVP-AUY922, STA-9090, AT13387, and KW-2478. Other inhibitors are synthetic ATP-competitive inhibitors developed through rational drug design and include purine-scaffold inhibitors (PUH71, MPC-3100, BIIB021, CUDC305, BIIB028), substituted-benzamides (SNX-5422, TAS116), tropane-containing benzamides (XL888), pyrimidine derivatives (NVP-HSP990), and two drugs for which structures have not been disclosed (DS2248, TQB3474). All these inhibitors target the NTD. Inhibitors interacting with the human Hsp90 CTD have also been described. After the coumarin-derived antibiotic novobiocin (NB), derivatives of natural compounds, including the green tea catechin Epigallocatechin-3-gallate (EGCG), the flavonolignan silybin (SB), and the rotenoid deguelin (DG), are currently evaluated in pre-clinical studies (Figure 1) [17,18,19,20,21].

In vitro studies on *Candida albicans* have highlighted the involvement of Hsp90 in major intracellular signaling pathways regulating resistance, apoptosis, cell wall integrity, morphogenesis, and virulence [22,23,24], making it a promising target for antifungal therapy. Several NTD-Hsp90 and some CTD-Hsp90 synthetic inhibitors have already been tested alone or in combination with azoles or echinocandins against a few *Candida* spp. in vitro and in a *Galleria mellonella* model [25,26,27,28,29,30,31,32,33,34,35] (Appendix A). Interestingly, efungumab (Mycograb^®^), a monoclonal antibody binding the middle domain of Hsp90, was in clinical trial as adjunctive therapy with amphotericin B in opportunistic fungal infections in the United States (registered as NCT00324025 and NCT00847678). At the same time, Mycograb^®^ has been demonstrated to have antitumor activity in vitro and was in phases I and II for breast cancer (registered as NCT00217815). Although it was granted orphan drug status by the Food and Drug Administration, the marketing authorization for Mycograb^®^ in the treatment of invasive candidiasis was refused by the European Medicine Agency in 2007 because the benefits did not outweigh the risks, and its development was stopped. These remarks and comments illustrate that the main challenge in the discovery of antifungal Hsp90 inhibitors is the high conservation of Hsp90 interspecies, reducing the therapeutic index for effective and safe antifungal drugs. More recently, resorcylate aminopyrazole and diarylisoxazole analogs were designed with potent fungal selectivity, supporting the feasibility of targeting fungal Hsp90 as a promising strategy [36,37].

Another challenge to consider when studying inhibitors with mechanisms of action already validated in humans is the difference in cell structure between *Candida* cells and human cells. One of the most remarkable phenotypical properties of fungal cells compared with animal cells is the presence of a cell wall surrounding the plasma membrane [38]. The length of mannan chains and the content of β-glucans and chitin influence the thickness and stiffness of this cell wall, affecting drug efficiency [39]. Composition and architecture modifications of the cell wall led to the emergence of less susceptible or resistant profiles [40,41]. The quantitative composition of the cell wall is also variable between *Candida* species, and structural features could explain some observed drug susceptibility differences [42,43]. Fungal Hsp90 is located predominantly in the yeast cytoplasm [44]. Targeting a cytosolic protein implies the use of drugs with size, lipophilicity, and solubility, allowing for passive diffusion through the plasma membrane or involving active carrier-mediated transport [45].

Comparing the activity of structurally unrelated Hsp90 inhibitors on the same *Candida* spp. strains and identifying the factors influencing this activity should allow us to contribute to antifungal drug discovery by determining which fungal Hsp90 inhibition strategy could be the most promising to develop more effective and selective compounds. Thus, we selected nine antitumor and antiproliferative drugs known to directly inhibit human Hsp90 activity by targeting the NTD or the CTD. To our knowledge, four of them have not yet been evaluated as antifungal drugs (SNX-5422, BIIB021, novobiocin, and deguelin). For each compound, in vitro antifungal activities were tested against one ATCC strain and the same panel of 26 clinical isolates (susceptible and resistant strains) representing the five most common species of *Candida*. The synergistic activity of the compounds was also evaluated against fluconazole and caspofungin-resistant strains. Finally, the physicochemical properties of the drugs, the Hsp90 amino acid sequences, and the residues involved in the interactions were compared to discuss factors that could influence both the inhibitory activity of the compounds and the new strategy to be developed in the framework of a drug discovery program.

## 2. Materials and Methods

### 2.1. Yeast Strains and Culture Conditions

All yeast strains were maintained at −80 °C in glycerol-supplemented medium for long-term storage or at 4 °C on Sabouraud–Chloramphenicol medium for short-term storage (no more than three months). The 26 *Candida* spp. Clinical isolates used in experiments were obtained from the Mycobank of the Parasitology and Medical Mycology Department, UR 1155, Nantes, France. All are clinical isolates with virulence and susceptibility already studied in the laboratory and permitted to constitute a panel representing the most common species (*C. albicans* susceptible or azole-resistant [46], *C. glabrata* susceptible or echinocandins resistant [47], *C. tropicalis*, *C. parapsilosis*, and *C. krusei*). The strains were identified by MALDI-TOF Mass Spectrometry.

### 2.2. Reagents

The Hsp90 inhibitor geldanamycin (ant-gl-5, >95%) was purchased from Invivogen, Toulouse, France. Radicicol (BR162744, >99%) was purchased from Carbosynth Ltd., Berkshire, UK. Deguelin (A13460, >98%), BIIB021 (A10143, >98%), and NVP-AUY922 (A10659, >98%) were purchased from Adooq Bioscience, Nanterre, France. Epigallocatechin-3-gallate (E4143, >95%) and silybin (S0417, >98%) were purchased from Sigma, Saint-Quentin-Fallavier, France. Novobiocin (BML-A256) was purchased from Enzo Life Science, Villeurbanne, France. All reagents were stored at −80 °C in DMSO (10 mM). Concerning the reference antifungal drug used in combination for the synergy assay, fluconazole (FD23320, >98%) and caspofungin (FC162691801, >98%) were purchased from Carbosynth Ltd., Berkshire, UK.

### 2.3. Antifungal Susceptibility Assay

Antifungal susceptibility testing was performed in flat bottom 96-well microliter plates with fluorochromic growth indicator resazurin assay as described previously [48]. Briefly, strains grew on Sabouraud–Chloramphenicol medium for 24 h at 37 °C. The inoculum suspensions were prepared in liquid RPMI 1640 medium (Sigma-Aldrich, Saint-Quentin-Fallavier, France) to a final concentration of 0.5 × 10^3^–2.5 × 10^3^ yeasts/mL. All Hsp90 inhibitors were prepared in supplemented (MOPS 0.33 M—glucose 4%—pH 7) RPMI 1640-L-glutamine medium and a final DMSO concentration of 1%. Each concentration of molecules (100 µL) to be tested was added (in triplicate), and plates were incubated at 37 °C for 24 h. The cellular viability was evaluated on the Fluorolite 1000 (Dynatech, Guyancourt, France) with an excitation at 550 nm and an emission at 590 nm after a 4 h incubation with 10 µL of resazurin. The half-maximal inhibitory concentration (IC_50_) is the concentration that inhibited 50% of the cell growth and was determined by linear regression analysis. IC_50_ was expressed as the mean of the triplicate values repeated twice.

### 2.4. Synergy Assay

Antifungal synergy was tested using resistant clinical isolates with molecular mechanisms already identified in the lab. Synergy with fluconazole was evaluated against a resistant *C. albicans* strain (ERG11 and ERG3 mutations, IC_50_ > 100 µg/mL). Synergy with caspofungine was evaluated against two resistant *C. glabrata* strains (FKS mutations, IC_50_ > 4 µg/mL). Hsp90 inhibitors were added to reference drugs at a concentration 5–10 times lower than their own IC_50_. For the calculation of the fractional inhibitory concentration index (FICI), values of IC_50_ were set so as not to overestimate the effect of the association. For example, IC_50_ > 100 UI was set as 101 UI, IC_50_ = 100 ± 50 UI was set as 50 UI, and IC_50_ < 1 UI was set as 0.9 UI.

### 2.5. Sequence Alignment and Inhibitor Binding Site Identification

The sequences of human Hsp90α (UniProtKB code P07900), Hsp83 of *Trypanosoma brucei* (UniProtKB code Q389P1), Hsp82 of the yeast *Saccharomyces cerevisiae* (UniProtKB code P02829) and Hsp90 of different *Candida* species (UniProtKB code P46598, C5MG15, G8BKN6, and Q6FLU9, for respectively *C. albicans*, *C. tropicalis*, *C. parapsilosis* and *C. glabrata*) were aligned using ClustalΩ and ESPript (https://espript.ibcp.fr, accessed on 5 April 2021) [49]. Based on the ligand–protein crystal structure listed in the Protein DataBank in Europe and data of literature describing the docking-based analysis, the Hsp90 residues supposed to interact with ATP and the different inhibitors were identified on the sequence alignments.

### 2.6. Physicochemical and ADME Properties Prediction

The partition coefficient (Log Po/w), water solubility (Log S), and theoretical pharmacokinetic (ADME) properties of the nine Hsp90 inhibitors tested were run using the SwissADME web tool (http://www.swissadme.ch/, accessed on 7 July 2021) to predict their potential fate in yeast in vitro. These predictions were compared with antifungal drugs already used in clinics (anidulafungin, caspofungin, micafungin, fluconazole, itraconazole, voriconazole, posaconazole, flucytosine, and amphotericin B) whose experimental properties were analyzed in the DrugBank database.

## 3. Results

### 3.1. Antifungal Activity of Human Hsp90 Inhibitors

The half-maximal inhibitory concentrations (IC_50_) of the nine Hsp90 inhibitors against 1 ATCC strain (SC5314/ATCC MYA-2876) and 26 clinical isolates representing the five most common *Candida* species are shown in Table 1.

#### 3.1.1. N-Terminal Domain Inhibitors

Regarding the NTD inhibitors, only the ansamycin GA displayed antifungal activity against all the species, with IC_50_ values ranging from 3.5 ± 1.2 µM to 33.1 ± 2.1 µM. As regards the reproducibility of literature data, GA was used as a reference to validate our experiments. RA, the second natural product which was the origin of numerous inhibitors, was also evaluated. RA was active against 5 of 9 *C. albicans* strains, 2 of 4 *C. parapsilosis* strains, and 1 of 6 *C. glabrata* strains (IC_50_ values between 25.2 ± 5.2 µM and 53.2 ± 13.8 µM) but remained inactive against *C. tropicalis* and *C. krusei* strains. Interestingly, its resorcinol derivative NVP-AUY922 showed a broader spectrum of activity than RA. Indeed, it was active against all tested *Candida* strains except *C. glabrata* (IC_50_ > 100 µM). NVP-AUY922 inhibited the growth of *C. albicans* (7/9), *C. krusei* (4/4), 3 of 4 *C. tropicalis* and *C. parapsilosis* strains with means of IC_50_ ranging from 18.0 ± 3.8 µM to 66.4 ± 14.2 µM. SNX-5422 and BIIB021 inhibitors, obtained by structure-based rational drug design strategy, were included in the evaluations. They are currently the most advanced in oncologic clinical trial phases. SNX-5422 did not show significant antifungal activity with weak growth inhibition in 2 of the 6 *C. albicans* strains (IC_50s_ between 51.8 ± 7.2 µM and 55.1 ± 6.1 µM) but none of the other 4 species. The purine scaffold BIIB021 was not active against any species at concentrations up to 100 µM.

#### 3.1.2. C-Terminal Domain Inhibitors

Concerning the CTD inhibitors, it is interesting to note that the green tea catechin EGCG showed activity against the five species. EGCG inhibited the growth of all the tested strains except *C. parapsilosis* (2 out of 4). The IC_50_ values of EGCG were lower than 1 µM up to 58.1 ± 17.7 µM. The in vitro activities were ten times higher against *C. glabrata* (IC50s between 0.82 ± 0.19 and 4.8 ± 3.6 µM) and *C. krusei* (IC_50s_ between 2.5 ± 0.7 and 5.3 ± 3.4 µM) than against *C. albicans* (IC_50s_ between 15.1 ± 11.7 and 58.1 ± 17.7 µM) and *C. tropicalis* (IC_50s_ between 22.3.1 ± 5.4 and 56.8 ± 14.8 µM). Three other natural products described as CTD inhibitors were tested, but none showed relevant activity. Silybin, deguelin, and novobiocin had no effect on strain proliferation at concentrations up to 100 µM.

#### 3.1.3. Combination with Fluconazole and Caspofungin

The nine inhibitors were also tested in combination with fluconazole against one *C. albicans*-resistant isolate and in combination with caspofungin against two *C. glabrata*-resistant isolates (Table 2). The fractional inhibitory concentration index (FICI) [50] was calculated for all combinations using the formula FICI = FICI(drug A) + FICI(drug B), where FICI(drug A) = [(IC_50_ drug A in combination)/(IC_50_ drug A alone)], same for drug B. The effects of the antifungal drug combinations were classified according to the following criteria: FICI ≤ 0.5: synergistic effect; 0.5 < FICI ≤ 1: additive effect; 1 < FICI < 4: no interaction; FICI ≥ 4.0: antagonistic effect [51]. Except for BIIB021, the combination of all the NTD inhibitors with fluconazole displayed a synergistic effect against the fluconazole-resistant *C. albicans* strain. Interestingly, the additional effect of caspofungin is also noticeable with three of the NTD inhibitors, including RA and NVP-AUY922, which had no effect alone against caspofungin-resistant *C. glabrata* strains. Regarding the CTD inhibitors, only the combination of EGCG with fluconazole or caspofungin (especially against the CAGL27 strain) displayed a synergistic effect. None of the nine Hsp90 inhibitors showed antagonism with fluconazole or caspofungin.

### 3.2. Physicochemical Properties of Human Hsp90 Inhibitors

Some of the Hsp90 inhibitors, which display cytotoxicity on mammalian cells, do not have direct antifungal activity or activity in combination with reference antifungal drugs. This difference in activity may be related to structural differences between human and fungal cells. Referring to the SwissADME web tool [52] for the prediction of physicochemical properties and druggability of the molecules, the compatibility of Hsp90 inhibitors with antifungal use was investigated (Table 3). These predictions were compared with the theoretical and experimental properties of the reference systemic antifungal drugs (echinocandins, azoles, amphotericin B, and flucytosine), all of which must penetrate the fungal cell wall to reach their target. The antifungal drugs crossing the cell wall have a molecular weight in the wide range of 129.09 to 1270.27 g·mol^−1^, an experimental LogP_o/w_ in the range of −1.1 to 5.66 (theoretical −2.98 to 4.74), and a heterogeneous solubility. The largest and least lipophilic chemical structures, such as echinocandins and amphotericin B, correspond to drugs reaching targets located in the plasma membrane (glucan synthase and ergosterol, respectively) and, therefore, do not need to penetrate the cytoplasm. In addition, drugs that need to cross the lipid bilayer to reach their target have a molecular weight lower than 705.63 g·mol^−1^ and mainly use a specific carrier protein to facilitate diffusion. The Hsp90 inhibitors tested have a molecular weight between 318.76 and 612.62 g·mol^−1^ and a theoretical LogP_o/w_ between 1.00 and 3.49. The mechanism of entry of Hsp90 inhibitors into cells is still not described, but even in the absence of a specific transporter, their physicochemical properties seem theoretically compatible with passive diffusion through fungal structures to reach their respective target.

### 3.3. Binding Sites of Human Hsp90 Inhibitors in Candida

As cellular or structural differences between humans and yeast do not theoretically explain the results for the non-effectiveness of the compounds against fungal cells, the molecular level was investigated.

#### 3.3.1. Multiple Sequences Alignment

To identify the molecular factors influencing the variability of Hsp90 inhibitor activity between humans and *Candida*, the sequences of Hsp90 homologs available on Uniprot.org were aligned (Figure 2). Multiple sequences alignment of human Hsp90α with *Candida* sp. Hsp90 homologs showed an amino acid identity of around 60%. This identity is higher when focusing on the NTD (67%), while alignment of the CTD interestingly displayed a human and yeast sequence identity of less than 50%. Furthermore, multiple sequence alignment of total Hsp90, NTD, and CTD of the different *Candida* species showed ranges of identity between 90.2% and 97.2%, except for *C. glabrata*, which, as expected by its taxonomic position, is more identical to *S. cerevisiae* than the other *Candida* species (Table 4). To highlight whether these differences relate to the residues involved in the interaction of the ligands with the protein, sequence alignment was performed and annotated with reference to previous published X-ray crystallography of protein (human and yeast)-ligand interaction and molecular docking studies (Figure 2). The co-crystal structures are referenced in the European Protein DataBank for all NTD inhibitors tested, with the exception of SNX-5422. As SNX-5422 is the SNX-2112 prodrug, it was considered that interactions could be similar, except for the residue K58 binding the hydroxyl, which is esterified to give the prodrug (Table 5). The results showed that all amino acids potentially involved in ligand interactions with Hsp90 ATP-NTD, except the D102, which is expected to interact with NVP-AUY922, are conserved between humans and yeast. Indeed, in *C. glabrata,* this aspartic acid is replaced by a glutamic acid (E87) but without affecting the interaction between the ligand and the protein because it has the same size and properties. X-ray crystallography has never been used to determine the protein–ligand interaction between BIIB021 and yeast Hsp90. However, as interactions with the *Trypanosoma brucei* Hsp83 homolog have been described, this sequence was further aligned, showing 100% identity of the amino acids involved (Figure 2). This does not explain the complete lack of activity of BIIB021 against any *Candida* isolate.

#### 3.3.2. Computational Models Comparison

Concerning CTD, inhibitor binding sites are not well elucidated or not explored. Based on different computational models, the amino acids likely to be involved in NB and SIL/hHsp90α interactions were listed, and a percentage identity of residues in *Candida* Hsp90 homologs was applied for each model proposed by the authors (Table 6). Three models of NB binding have been indexed, and the percentage identity of residues varies between 12.5 and 100%. The model of Cuyàs et al. proposed several possibilities clustered around the putative binding site. In this study, two clusters differed if the ligand Root-Mean-Square Deviation of their atomic positions was higher than a minimum of 6 Å around the hot spot conformations. Only the key residues of Hsp90 interacting with the best ligand in each cluster were detected. Thus, they detected three clusters for the interaction between NB and Hsp90α closed conformation and four clusters for the interaction with the open conformation (four or six between SIL and closed or open Hsp90α conformations, respectively). Some of the clusters detected do not bind the CTD. Indeed, it should be noticed that cluster 3 on the hHsp90α-closed conformation implies interaction with the MD and that cluster 1 on the open conformation implies interaction with the NTD. Furthermore, cluster 2 on the open conformation was not considered as it involved only linker residues. Concerning the SIL bonds, the homology model of Cuyàs et al. also proposed numerous clusters, among which clusters 5 and 6 in the hHsp90α-closed conformation imply binding to the MD, and cluster 1 on the open conformation supposes bonds with the NTD. Considering all conformations, the percentage identity of homologous human and yeast amino acids that bind with SIL ranged from 38 to 80%. All amino acids that differ between human and *Candida* Hsp90 CTD are listed in Table 7, and those with different sizes and physicochemical properties have been highlighted in bold. These bolded amino acids could explain the difference in activity of the CTD inhibitors against human and fungal cell lines by decreasing the ligand binding. Thus, 11 amino acids were identified as being of potential interest for targeting by new compounds designed for CTD yeast specificity (human numbering: K560, K582, N590, T594, T607, N622, M625, A629, A630, Q682, and N686; *C. albicans* homologs respectively: A536, Q558, Y566, D570, S583, T598, S601, S605, S606, S659, H665). Focusing on the differences between Hsp90 *Homo sapiens* CTD (pDB IF: 7L7I) [66] and *Candida albicans* Hsp90 model structure obtained from AlphaFold [67,68] for prediction (black box in Figure 2), the superposition of the two structures (Figure 3) shows only slight variations in terms of structure while the nature of amino acids could be crucial for inhibitor binding. Models for the interactions of EGCG and DG with Hsp90 are not yet available.

## 4. Discussion

Targeting the Hsp90 as an alternative mechanism of action for the development of innovative antifungal drugs is no longer a concept to be proven. Several strategies could be used to inhibit the activity of this chaperone. The most common strategy is the inhibition of its ATPase activity by competing with the ATP-binding site located on the NTD. Among these NTD inhibitors, we confirmed that the old natural benzoquinone, geldanamycin, remains the most active and exhibits the broadest-spectrum antifungal activity against the most clinically relevant *Candida* species. However, both this molecule and its derivatives, tanespimycin (17-AAG) and alvespimycin (17-DMAG), have demonstrated poor pharmaceutical properties [73]. The resorcinolic isoxazole amine NVP-AUY922 also demonstrated significant activity against most *Candida* strains available in the laboratory, but approximately 80% of patients reported ocular side effects in clinical trials for cancer pathologies [74]. The other two NTD inhibitors, BIIB02 and SNX-5422, which have never been tested for antifungal activity, are not effective. A second ATP-binding site is located on the CTD. The CTD is the site of Hsp90 dimerization and client protein binding. Its nucleotide-binding site acts as an allosteric regulator of the NTD ATPase activity [75]. In our in vitro conditions, the EGCG inhibitor exhibited high intrinsic antifungal activity and synergistic activity with reference drugs against resistant clinical isolates. Data suggest that the binding of EGCG to Hsp90 impairs the association of Hsp90 with its co-chaperones, thereby inducing the degradation of Hsp90 client proteins, resulting in anti-proliferating effects in pancreatic cancer cells [76]. However, EGCG has dozens of molecular targets in oncology [21,77], and it is not possible to ensure, with the available data, that its antifungal activity is solely due to the inhibition of *Candida* Hsp90 CTD. None of the other CTD inhibitors tested were active against any *Candida* species. Overall, our results showed great variability in the antifungal activities of human Hsp90 inhibitors.

To understand the source of these differences in activity against yeast and human cells, we explored cellular and molecular hypotheses. With the available data and prediction of the Hsp90 inhibitor’s physicochemical properties, cell wall permeability did not appear to be a barrier. However, efflux mechanisms have been reported for Hsp90 in oncology and cannot be excluded. The resistant *C. albicans* isolate CAAL28 carried gain-of-function mutations in the MRR1 and TAC1 transcription factor genes [78]. These mutations are associated with overexpression of the P-glycoprotein (P-gp) efflux transporters Mdr1 and Cdr1. Resistance to Hsp90 inhibitors has been previously described as being linked to P-gp expression. This mechanism has been described as relevant for benzoquinone derivatives [79,80]. Previous studies have reported that Hsp90 inhibitors also modulate P-gp activity [81,82,83]. In this study, only geldanamycin and EGCG were active against the CAAL28 isolate. The lack of activity of other inhibitors could be related to active efflux.

Concerning the amino acid sequence of the target protein, the NTD of fungal and human Hsp90 is highly conserved, and the non-identical amino acids do not seem to correspond to residues that would elucidate human versus fungal specificity. Therefore, one might have imagined that all NTD inhibitors would be equally active against *Candida*. In fact, the three-dimensional folding of the target and its environment must be taken into account. Marcyk et al. published in 2021 the x-ray crystallographic structures of three fungal selective Hsp90 inhibitors, namely resorcylate aminopyrazoles (RAPs), in complex with the nucleotide-binding domain (PDB ID 7K9U, 7K9V, and 7K9W) of *Cryptococcus neoformans* Hsp90 [36]. Indeed, the authors explained this selectivity by conformational rearrangements induced by ligand binding that occur exclusively with the fungal proteins and not with its human homolog. As NTD targeting does not seem to be the most obvious strategy to develop new antifungal inhibitors, the differences in protein flexibility could lead to the development of more selective drugs.

With regard to CTD, targeting this domain is beginning to be explored in oncology with covalent inhibitors [84]. They showed a disruption of the in vitro function of human Hsp90 in mammalian cells, whereas we did not find any reproducibility of this activity against fungal cells. In contrast to what was observed for NTD, the low identity of CTD sequences between humans and yeast could alone explain the lack of binding of the tested inhibitors to the fungal protein. DDO-6600, identified as a covalent inhibitor, interacts with Cys598 of Hsp90, which corresponds to an alanine in yeast [84]. Furthermore, based on previously published docking studies [70,71,72] and our computational model, the residues likely to be responsible for specific binding in human Hsp90 were identified, along with the corresponding amino acids involved in *Candida*. The hypotheses explored in this work could serve as a launching point for future virtual screening studies based on the amino acids highlighted. Targeting these amino acids could be an interesting strategy to develop or identify new and more selective antifungal compounds.

## Figures and Tables

**Figure 1 microorganisms-11-02837-f001:**
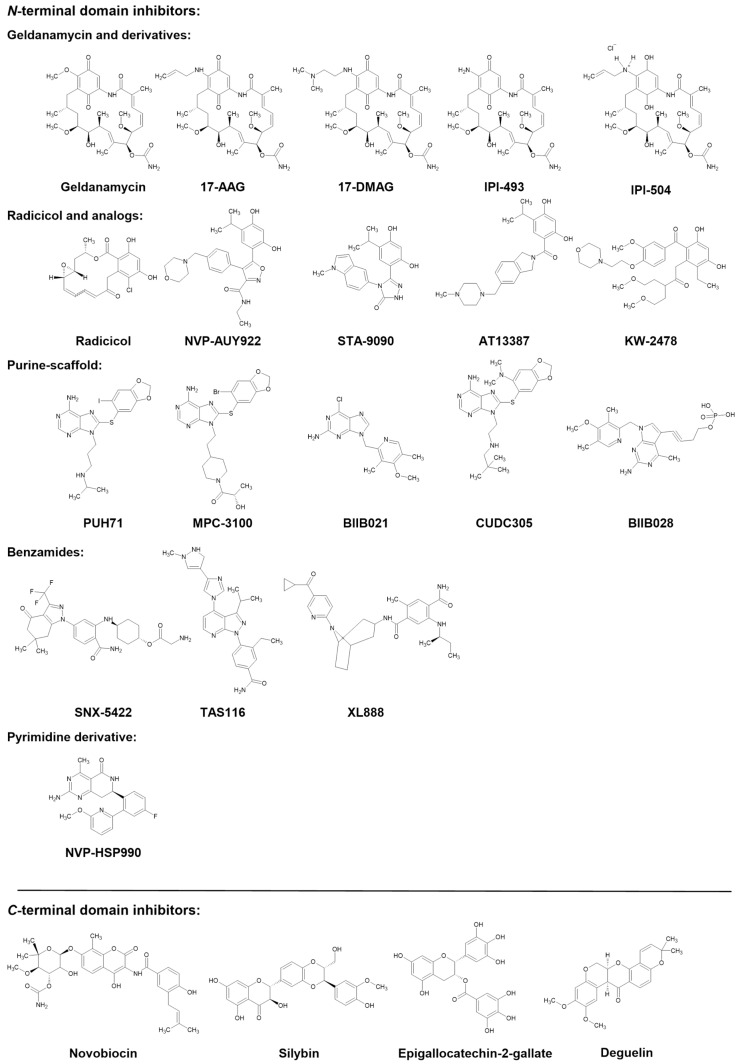
Chemical structures of the Hsp90 inhibitors.

**Figure 2 microorganisms-11-02837-f002:**
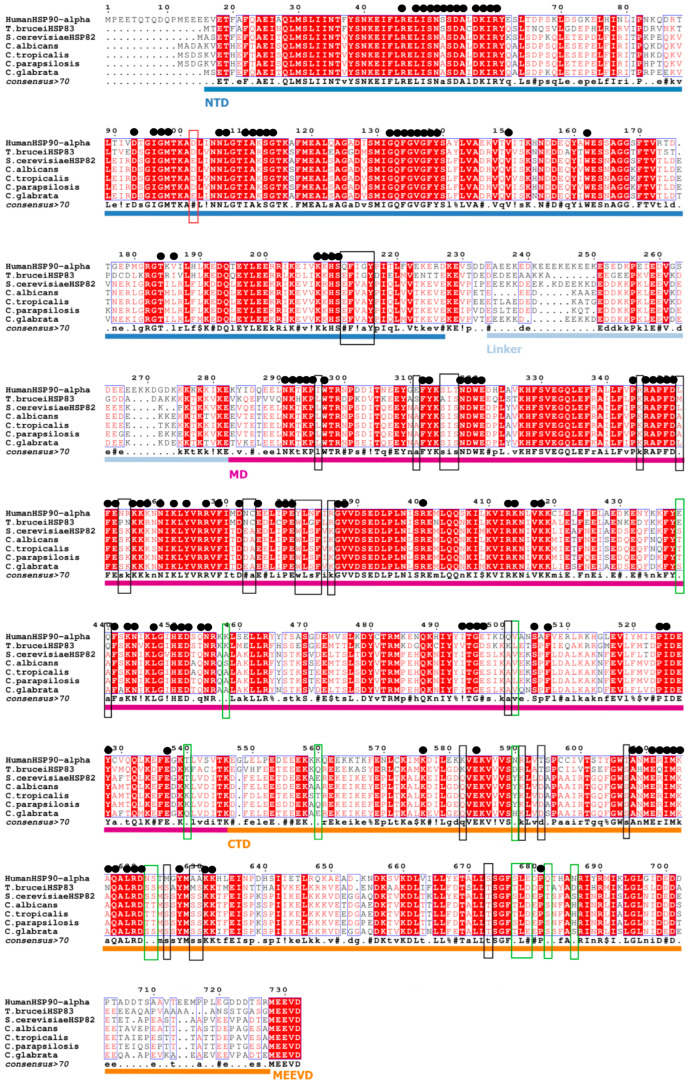
Multiple sequences alignment of human Hsp90 *alpha*, *Trypanosoma brucei* Hsp83, *Saccharomyces cerevisiae*. Boxes represent residues or sequences of residues susceptible to interaction with Hsp90 inhibitors and different between humans and *Candida* sp. (black), different between the four *Candida* species (green), or different only for one *Candida* species (red). The black dots represent residues or sequences of residues susceptible to interaction with ATP and/or Hsp90 inhibitors and are identical between humans and yeast. NTD, *N*-Terminal Domain; MD, Median Domain; CTD, *C*-Terminal Domain.

**Figure 3 microorganisms-11-02837-f003:**
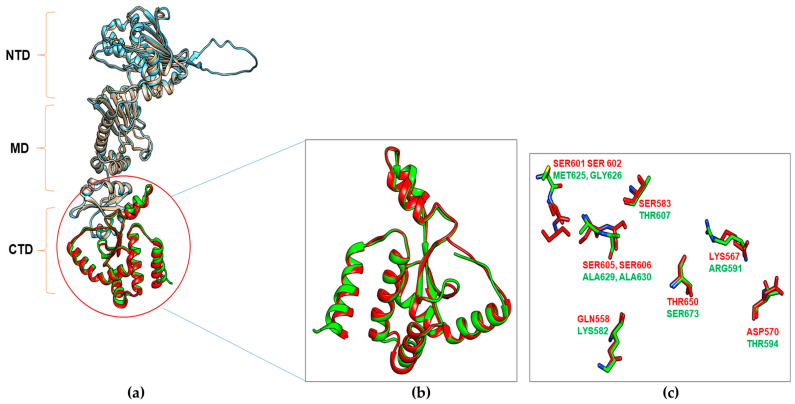
Structure superposition of HSP90 *Homo sapiens* (pdb id: 7L7I) with *Candida albicans* HSP90 model obtained from AlphaFold prediction. (**a**) Three domains, the N-terminal domain (NTD), Median domain (MD), and C-terminal domain (CTD), are represented. (**b**) Superimposed structure of C-terminal domain of HSP90 of *Homo sapiens* (Red) and *Candida albicans* (Green). The RMSD value between both structures is ~1.0 Å. (**c**) *Homo sapiens* and *C. albicans* residues susceptible to interact with HSP90 inhibitors.

**Table 1 microorganisms-11-02837-t001:** Antifungal in vitro activity of the Hsp90 inhibitors against SC5314 and *Candida* clinical isolates.

Species	Isolates	IC_50_ (µM)									
		VRC	GA	RA	NVP-AUY922	SNX-5422	BIIB021	EGCG	SIL	DG	NB
*C. albicans*	SC5314	<0.25	24.0 ± 6.2	27.5 ± 2.9	18.0 ± 3.8	>100	>100	18.0 ± 3.8	>100	>100	>100
	CAAL118	<0.25	28.6 ± 5.3	>100	34.4 ± 13.2	>100	>100	35.3 ± 10.7	>100	>100	>100
	CAAL16	<0.25	20.3 ± 1.2	32.4 ± 7.5	36.0 ± 6.9	>100	>100	32.9 ± 5.5	>100	>100	>100
	CAAL93	<0.25	17.9 ± 7.1	25.2 ± 5.3	34.4 ± 4.1	>100	>100	15.1 ± 11.7	>100	>100	>100
	CAAL97	<0.25	29.8 ± 0.7	37.4 ± 3.4	37.7 ± 3.2	51.8 ± 7.2	>100	29.1 ± 7.5	>100	>100	>100
	CAAL2	>25	3.5 ± 1.2	49.4 ± 8.5	47.3 ± 15.2	55.1 ± 6.1	>100	58.1 ± 17.7	>100	>100	>100
	CAAL28	>25	24.9 ± 9.3	>100	>100	>100	>100	48.8 ± 4.9	>100	>100	>100
	CAAL111	ND	23.3 ± 2.2	>100	>100	>100	>100	30.4 ± 8.1	>100	>100	>100
	CAAL117	ND	25.3 ± 0.9	>100	38.5 ± 7.9	>100	>100	28.6 ± 10.3	>100	>100	>100
Mean ± SD		-	14.0 ± 11.3	-	-	-	>100	35.4 ± 15.7	>100	>100	>100
*C. glabrata*	CAGL1	7.10 ± 2.28	>10	>100	>100	>100	>100	2.9 ± 1.1	>100	>100	>100
	CAGL2	<0.25	>10	>100	>100	>100	>100	3.3 ± 2.1	>100	>100	>100
	CAGL3	<0.25	>10	>100	>100	>100	>100	3.8 ± 1.5	>100	>100	>100
	CAGL4	<0.25	>10	53.1 ± 13.8	>100	>100	>100	4.8 ± 3.6	>100	>100	>100
	CAGL22	ND	6.4 ± 0.3	>100	>100	>100	>100	1.59 ± 1.45	>100	>100	>100
	CAGL27	ND	10.6 ± 4.5	>100	>100	>100	>100	0.82 ± 0.19	>100	>100	>100
Mean ± SD		-	-	-	>100	>100	>100	3.12 ± 2.34	>100	>100	>100
*C. tropicalis*	CATR1	<0.25	29.1 ± 2.1	>100	>100	>100	>100	22.3 ± 5.4	>100	>100	>100
	CATR2	<0.25	33.1 ± 2.1	>100	66.4 ± 14.2	>100	>100	31.6 ± 6.3	>100	>100	>100
	CATR3	2.79 ± 0.25	32.3 ± 2.9	>100	34.9 ± 4.3	>100	>100	52.0 ± 14.2	>100	>100	>100
	CATR4	25.4 ± 5.33	29.0 ± 4.9	>100	30.9 ± 3.9	>100	>100	56.8 ± 14.8	>100	>100	>100
Mean ± SD		-	30.9 ± 3.6	>100	-	>100	>100	39.0 ± 14.8	>100	>100	>100
*C. parapsilosis*	CAPA1	7.61 ± 3.04	21.6 ± 3.1	>100	42.5 ± 6.8	>100	>100	15.5 ± 5.0	>100	>100	>100
	CAPA17	<0.25	11.7 ± 9.4	32.7 ± 2.3	30.3 ± 1.6	>100	>100	>100	>100	>100	>100
	CAPA2	5.33 ± 2.79	9.6 ± 5.6	>100	>100	>100	>100	18.1 ± 8.0	>100	>100	>100
	CAPA3	<0.25	10.4 ± 8.3	30.8 ± 5.0	25.8 ± 3.8	>100	>100	>100	>100	>100	>100
Mean ± SD		-	12.1 ± 8.0	-	-	>100	>100	-	>100	>100	>100
*C. krusei*	CAKR1	<0.25	26.2 ± 2.2	>100	30.5 ± 2.8	>100	>100	3.9 ± 0.9	>100	>100	>100
	CAKR2	<0.25	>10	>100	28.8 ± 10.2	>100	>100	2.5 ± 0.7	>100	>100	>100
	CAKR3	<0.25	22.3 ± 7.6	>100	35.4 ± 5.4	>100	>100	3.0 ± 0.5	>100	>100	>100
	CAKR4	0.46 ± 0.00	27.8 ± 4.2	>100	32.5 ± 2.3	>100	>100	5.3 ± 3.4	>100	>100	>100
Mean ± SD		-	-	>100	32.2 ± 5.2	>100	>100	3.8 ± 2.1	>100	>100	>100

Values represent the means of IC_50_ ± SD (µM) of the experiment performed at least twice in triplicate. VRC, Voriconazol; GA, Geldanamycin; RA, Radicicol; EGCG, Epigallocatechin-3-gallate; SIL, Silybin; DG, Deguelin; NB, Novobiocin; IC_50_, half maximal inhibitory concentration.

**Table 2 microorganisms-11-02837-t002:** Antifungal in vitro activity of the Hsp90 inhibitors in combination with reference drugs against resistant *Candida* clinical isolates.

Isolates	Species	IC_50_										
		FLC ^a^	FLC/GA ^b^	FICI ^c^	FLC/RA ^b^	FICI	FLC/AUY922 ^b^	FICI	FLC/SNX-5422 ^b^	FICI	FLC/EGCG ^b^	FICI
*C. albicans*	CAAL2	101	0.9/1	0.44	0.9/10	0.25	0.9/10	0.32	0.9/10	0.21	0.9/10	0.26
		CSP ^a^	CSP/GA ^b^	FICI ^c^	CSP/RA ^b^	FICI	CSP/AUY922 ^b^	FICI	CSP/SNX5422 ^b^	FICI	CSP/EGCG ^b^	FICI
*C. glabrata*	CAGL22	5	0.18/10	1.8	0.24/100	1.1	0.74/100	1.3	5/101	NI	0.68/0.5	3.8
	CAGL27	5	0.20/10	1.8	0.24/100	1.1	0.63/100	1.5	5/101	NI	0.23/0.25	0.46

^a^ Values represent the IC_50_ for FLC and CSP alone (µg/mL); ^b^ values represent IC_50_ (µg/mL) for FLC or CSP and IC_50_ (µM) for GA, RA, AUY922, SNX5422, or EGCG in combination, respectively; ^c^ values represent the FICI values. IC_50_, half maximal inhibitory concentration; ND, not determined; NI, no inhibition at the maximal concentration of Hsp90 inhibitors (100 µM) and FLC (100 µg/mL) or CSP (4 µg/mL); FICI, Fractional Inhibitory Concentration Index; FLC, fluconazole; CSP, caspofungine.

**Table 3 microorganisms-11-02837-t003:** Physicochemical predicted properties of antifungal reference drugs (highlighted in grey) compared with Hsp90 inhibitors.

Compounds	Target Location	Mechanism for Drug Entry in Cell	MW ^a^	Log P_o/w_ ^b^	Log S ^c^	Drug-Likeness ^d^
Micafungin	Cell membrane	Extracellular target	1270.27	−3.04 (exp.: 0)	−6.63 ± 1.86 poorly soluble	No
Caspofungin	Cell membrane	Extracellular target	1093.31	−1.82 (exp.: 0)	−6.38 ± 1.89 poorly soluble	No
Amphotericin B	Cell membrane	Extracellular target	924.08	−0.39 (exp.: 0.8)	−2.88 ± 5.11 soluble	No
Anidulafungin	Cell membrane	Extracellular target	1140.24	−0.34 (exp.: 2.9)	−8.57 ± 1.20 poorly soluble	No
Flucytosine	Nucleus	Purine/cytosine permease Fcy2 and homologous [53]	129.09	0 (exp.: −1.1)	−0.73 ± 0.65 very soluble	Yes
Fluconazole	Endoplasmic reticulum	Azole-specific energy-independent facilitated diffusion [54]	306.27	0.88 (exp.: 0.5)	−2.45 ± 0.98 soluble	Yes
Voriconazole	Endoplasmic reticulum	Azole-specific energy-independent facilitated diffusion [54]	349.31	2.40	−3.77 ± 1.46 soluble	Yes
Posaconazole	Endoplasmic reticulum	Diffusion into the plasma membrane but mechanism undefined [55]	700.78	3.33 (exp.: 5.5)	−7.33 ± 1.08 poorly soluble	No
Itraconazole	Endoplasmic reticulum	Azole-specific energy-independent facilitated diffusion [54]	705.63	4.71 (exp.: 5.66)	−8.11 ± 0.98 poorly soluble	No
EGCG	Cell wall Cytoplasm	Unidentified	458.37	1.01	−3.66 ± 1.21 soluble	No
GA	Cell wall Cytoplasm	Unidentified	560.64	1.57	−4.18 ± 0.90 moderately soluble	No
AUY922	Cell wall Cytoplasm	Unidentified	465. 54	3.36	−5.81 ± 1.24 moderately soluble	Yes
RA	Cell wall Cytoplasm	Unidentified	364.78	2.51	−4.12 ± 1.11 moderately soluble	Yes
SNX-2112	Cell wall Cytoplasm	Unidentified	464.48	3.49	−5.54 ± 0.42 moderately soluble	Yes
SNX-5422	Cell wall Cytoplasm	Unidentified	521.53	3.13	−5.70 ± 0.64 moderately soluble	No (prodrug)
BIIB021	Cell wall Cytoplasm	Unidentified	318.76	1.81	−3.95 ± 0.94 soluble	Yes
SIL	Cell wall Cytoplasm	Unidentified	482.44	1.92	−4.47 ± 0.32 moderately soluble	Yes

Properties were predicted using the SwissADME web tool. ^a^ molecular weight (g·mol^−1^); ^b^ average consensus computed by SwissADME based on five predictions (iLOGP, XLOGP3, WLOGP, MLOGP, SILICOS-IT); ^c^ average of the three models of estimation (ESOL, Ali, SILICOS-IT), insoluble < −10 < poorly < −6 < moderately < −4 < soluble < −2 < very < 0 < highly; ^d^ Lipinski’s rule of five: MW≤ 500 g·mol^−1^, LogP ≤ 5, H-bond donor ≤ 5 and H-bond acceptor.

**Table 4 microorganisms-11-02837-t004:** Percentage of amino acid identity in the heat shock protein 90 sequences between the four different *Candida* species, human and *S. cerevisiae*.

Identity (%)	Human α	*C. albicans*	*C. tropicalis*	*C. parapsilosis*	*C. glabrata*	*S. cerevisiae*
Hsp90 total						
Human α		60.5	60.2	60.6	59.5	59.8
*C. albicans*	60.5		95.1	92.4	82.5	82.7
*C. tropicalis*	60.2	95.1		93.2	82.4	82.7
*C. parapsilosis*	60.6	92.4	93.2		82.4	83.5
*C. glabrata*	59.5	82.5	82.4	82.4		90.7
*S. cerevisiae*	59.8	82.7	82.7	83.5	90.7	
*N*-terminal domain						
Human α		67.7	67.2	66.8	65.9	65.1
*C. albicans*	67.7		97.2	96.8	85.8	85.8
*C. tropicalis*	67.2	97.2		96.3	86.2	86.2
*C. parapsilosis*	66.8	96.8	96.3		86.7	86.7
*C. glabrata*	65.9	85.8	86.2	86.7		92.6
*S. cerevisiae*	65.1	85.8	86.2	86.7	92.6	
*C*-terminal domain						
Human α		48.9	48.7	49.5	48.9	50.0
*C. albicans*	48.9		93.4	90.2	77.6	79.2
*C. tropicalis*	48.7	93.4		90.2	78.1	78.1
*C. parapsilosis*	49.5	90.2	90.2		75.4	77.6
*C. glabrata*	48.9	77.6	78.1	75.4		89.0
*S. cerevisiae*	50.0	79.2	78.1	77.6	89.0	

Values represent the percentage of amino acid identity determined after alignment from data available on the UniProt database.

**Table 5 microorganisms-11-02837-t005:** Percentage of identity between the residues involved in the interaction of the ligand and Hsp90-NTD between humans and yeast.

Ligand	Methods for Investigate the Interaction	Protein	PDB Code	% Identity of Residues Involved in the Drug–Protein Interaction	Ref.
ATP	X-ray diffraction	*S. cerevisiae* Hsp82	2CG9	100%	[56]
		Human Hsp90 alpha	3T0Z		[57]
GA	X-ray diffraction	Human Hsp90 alpha	1YET	100%	[58]
		*S. cerevisiae* Hsp82	2YGF		[59]
RA	X-ray diffraction	*S. cerevisiae* Hsp82	1BGQ	100%	[60]
		Human Hsp90 alpha	4EGK		[61]
		*C. albicans* Hsp90	6CJL		[27]
AUY922	X-ray diffraction	Human Hsp90 alpha	2VCI	95%	[62]
		*C. albicans* Hsp90 homolog	6CJS		[27]
SNX-5422					
SNX-2112	X-ray diffraction	Human Hsp90 alpha	4NH7	100%	[63]
		*C. albicans* Hsp90 homolog	6CJR		[27]
BIIB021	X-ray diffraction	*T. brucei* Hsp83	3O6O	100%	[64]
		Human Hsp90 alpha	3QDD		[65]

Percentages were determined based on X-ray crystallography data from the European Protein DataBank, compared with *Candida* sequences aligned from the UniProt database.

**Table 6 microorganisms-11-02837-t006:** Percentage of identity between the residues involved in the interaction of the ligand and Hsp90-CTD between humans and yeast.

Ligand	Methods for Investigating the Interaction	Computational Model	% Identity of Residues Involved in the Drug–Protein Interaction		Ref.
ATP	Homology model (*h*Hsp90α)	Sgobba et al., 2008	87%		[69]
NB	Homology model (*h*Hsp90α) Molecular docking	Sgobba et al., 2010	Cluster1 62.5% Cluster2 75%		[70]
		Matts et al., 2011	12.5%		[71]
		Cuyàs et al., 2019	Closed conformation: Cluster1 74% Cluster2 92% Cluster3 67%	Open conformation: Cluster1 85% Cluster3 100% Cluster4 62%	[72]
SIL	Homology model (*h*Hsp90α) Molecular docking	Cuyàs et al., 2019	Closed conformation: Cluster1 75% Cluster2 64% Cluster3 77% Cluster4 38% Cluster5 80% Cluster6 76%	Open conformation: Cluster1 77% Cluster2 79% Cluster3 75%	[72]
DG	Not investigated				
EGCG	Not investigated				

Percentages were determined based on molecular docking on the homology model from literature data, compared with *Candida* sequences aligned from the UniProt database.

**Table 7 microorganisms-11-02837-t007:** Amino acid specific to the interaction with the CTD domain of *h*Hsp90α versus *Candida* homolog.

Position (*h*Hsp90α)	aa		*Candida* Species Concerned	Nature		pH-Neutral Charge		MW		Ligand Binding	Ref.
	(1)	(2)		(1)	(2)	(1)	(2)	(1)	(2)		
**560**	**K**	**A**	* **C. albicans** *	**Basic**	**Aliphatic**	**(+)**	**uncharged non-polar**	**146.19**	**89.09**	NB	[71]
		**T**	* **C. tropicalis** *	**Basic**	**Hydroxylic**	**(+)**	**uncharged polar**	146.19	119.12		
		**Q**	* **C. parapsilosis** *	**Basic**	**Amide**	**(+)**	**uncharged polar**	146.19	146.15		
		**E**	* **C. glabrata** *	**Basic**	**Acid**	**(+)**	**(−)**	146.19	147.13		
**582**	**K**	**Q**	***Candida* sp.**	**Basic**	**Amide**	**(+)**	**uncharged polar**	146.19	146.15	NB	[70]
**590**	**N**	**Y**	** *C. albicans, C. tropicalis, C. parapsilosis* **	**Amide**	**Aromatic**	uncharged polar	uncharged polar	132.12	181.19	NB	[71]
		**H**	** *C. glabrata* **	**Amide**	**Basic**	**uncharged polar**	**(+)**	132.12	155.16		
591	R	K	*Candida* sp.	Basic	Basic	(+)	(+)	174.20	146.19	NB	[71]
**594**	**T**	**D**	***Candida* sp.**	**Hydroxylic**	**Acid**	**uncharged non-polar**	**(−)**	119.12	133.10	NB	[71]
**607**	**T**	**S**	***Candida* sp.**	Hydroxylic	Hydroxylic	**uncharged non-polar**	**uncharged polar**	119.12	105.09	NB	[70]
**622**	**N**	**T**	** *C. albicans, C. tropicalis, C. parapsilosis* **	**Amide**	**Hydroxylic**	uncharged polar	uncharged polar	132.12	119.12	NB	[72]
		**S**	** *C. glabrata* **	**Amide**	**Hydroxylic**	uncharged polar	uncharged polar	132.12	105.09		
623	S	T	*C. albicans, C. tropicalis, C. parapsilosis*	Hydroxylic	Hydroxylic	uncharged polar	uncharged polar	105.09	119.12	NB et SIL	[72]
**625**	**M**	**S**	***Candida* sp.**	**Sulphuric**	**Hydroxylic**	**uncharged non-polar**	**uncharged polar**	**149.21**	**105.09**	NB	[72]
**629**	**A**	**S**	***Candida* sp.**	**Aliphatic**	**Hydroxylic**	**uncharged non-polar**	**uncharged polar**	89.09	105.09	NB	[70,72]
**630**	**A**	**S**	***Candida* sp.**	**Aliphatic**	**Hydroxylic**	**uncharged non-polar**	**uncharged polar**	89.09	105.09	NB	[72]
673	S	T	*Candida* sp.	Hydroxylic	Hydroxylic	uncharged polar	uncharged polar	105.09	119.12	SIL	[72]
677	S	T	*C. albicans, C. tropicalis, C. glabrata*	Hydroxylic	Hydroxylic	uncharged polar	uncharged polar	105.09	119.12	ATP	
679	E	D	*C. albicans, C. tropicalis, C. parapsilosis*	Acid	Acid	(−)	(−)	147.13	133.10	ATP, NB	[70]
**682**	**Q**	**S**	***Candida* sp.**	**Amide**	**Hydroxylic**	uncharged polar	uncharged polar	146.15	105.09	NB	[71]
**686**	**N**	**H**	** *C. albicans, C. tropicalis, C. parapsilosis* **	**Amide**	**Basic**	**uncharged polar**	**(+)**	132.12	155.16	NB	[71]
		**S**	** *C. glabrata* **	**Amide**	**Hydroxylic**	uncharged polar	uncharged polar	132.12	105.09		

(1) Name, nature, physicochemical properties, and molecular weight of the amino acid (aa) present in the *h*Hsp90α sequence. (2) name, nature, physicochemical properties, and molecular weight of the amino acid (aa) present in the *Candida* Hsp90 homolog sequence. *h*Hsp90α, human Hsp90-alpha; (+), positively charged; (−), negatively charged; MW, Molecular Weight (g·mol^−1^).

## Data Availability

The datasets used and/or analyzed during the current study are available from the corresponding author on request.

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
