# Peer review of "Is the C-Terminal Domain an Effective and Selective Target for the Design of Hsp90 Inhibitors against Candida Yeast?"

_microorganisms, 2023, doi:10.3390/microorganisms11122837_

Round 1

Reviewer 1 Report

Comments and Suggestions for Authors

Pape et al. reported the antifungal activity of Hsp90 inhibitors against a variety of Candida clinical isolates, and compared the structural differences of hHsp90 and fungal Hsp90. While these studies are interesting, much work is needed to improve this manuscript and conduct supporting experiments first.

1. The INTRODUCTION part is too lengthy, and it should be shortened. Too many words are used to describe biological functions and inhibitors of Hsp90. On the contrary, some new reports on developing species-selective targeting inhibitors of fungal Hsp90 should be described in this part. For example, J. Med. Chem. 2022, 65, 7, 5539–5564.

2. Table should have a caption.

3. In Table 1, it is strange that the superior limit of IC50 is set to 10 μM for GA against C. glabrata, but 100 μM in other cases.

4. EGCG is a well-known multi-target compound, regulating a variety of biological macromolecules and signalings. Readers may be curious about the contribution of HSP90 inhibition into the antifungal or antitumor activity of EGCG. Some related discussions should be added.

5. Although the precise binding site of Hsp90 C-terminal inhibitors is not very clear, some reports have revealed that current C-terminal inhibitors may bind to different pockets. Does this result in the difference of antifungal activity between EGCG and other C-terminal inhibitors? Authors can add some discussions based on the structural analysis.

6. In Table 2, some IC50 values are zero?? By the way, "50" should be  subscript.

7. Prediction of physico-chemical properties does not add to the quality of the results and should be removed, unless authors can explain how these properties affect the antifungal activity of these Hsp90 inhibitors?

8. In addition to sequence alignment and conformation comparison, it would be nice to provide some molecular docking results to show how the sequence differences affect the Hsp90-inhibitor binding. It would be nicer if the molecular docking result could explain the activity differences of the tested inhibitors in Table 1.

9. The conclusion “the identification of selective CTD inhibitors of fungal Hsp90 could be a promising strategy for the development of innovative antifungal drugs” is based on “the greater sequence divergence in this domain”. However, the biological evaluation revealed that “the inhibitors binding to the C-terminal domain (CTD) did not show any antifungal activity, except one of them (ECGG, the multi-target one). ” Authors should indicate how to develop effective and species-selective antifungal Hsp90 inhibitors and provide a possible strategy in the manuscript.

10. There is no CONCLUSION part in this manuscript.

11. There are many format errors in this manuscript.

Author Response

Dear reviewer,

We appreciate your thorough review and the valuable comments you provided. In order to address a comment from the second reviewer, we conducted an additional experiment (IC50 of voriconazole, as presented in Table 1). Unfortunately, this supplementary experiment has caused a delay in our response. We apologize for any inconvenience this may have caused. Thank you for your understanding.

To follow up on your comments:

  1. The INTRODUCTION part is too lengthy, and it should be shortened. Too many words are used to describe biological functions and inhibitors of Hsp90. On the contrary, some new reports on developing species-selective targeting inhibitors of fungal Hsp90 should be described in this part. For example, J. Med. Chem.2022, 65, 7, 5539–5564.

In the introduction section, the mechanisms of action and biological pathways were less detailed and summarized (lines 65-71, 74-76, 89-93), have been removed from the text). References to recent reports on the design of selective analogs for fungal Hsp90 were mentioned (line109).

  1. Table should have a caption.

Shorter titles have been assigned to the tables. The data explanations and captions have been moved beneath the tables.

  1. In Table 1, it is strange that the superior limit of IC50 is set to 10 μM for GA against C. glabrata, but 100 μM in other cases.

These manipulations were not performed at the same time as the subsequent ones. In the course of our experimental approach, the evaluation began with a search for significantly active molecules and lower IC50 values. Then, given the results obtained with EGCG on these clinical isolates, we had not extended the exploration beyond concentrations greater than 10 µM for these experiments.

  1. EGCG is a well-known multi-target compound, regulating a variety of biological macromolecules and signalings. Readers may be curious about the contribution of HSP90 inhibition into the antifungal or antitumor activity of EGCG. Some related discussions should be added.
  2. Although the precise binding site of Hsp90 C-terminal inhibitors is not very clear, some reports have revealed that current C-terminal inhibitors may bind to different pockets. Does this result in the difference of antifungal activity between EGCG and other C-terminal inhibitors? Authors can add some discussions based on the structural analysis.

A reference to a report in oncology was mentioned in the discussion (line 438, ref 76). Ongoing research is being conducted within our research team regarding the interactions with EGCG in the fungus. We would like to refrain from mentioning them for a future publication.

  1. In Table 2, some IC50values are zero?? By the way, "50" should be  subscript.

The decimal transposition error from the Excel file has been corrected. All 50 of the IC50 values have been indexed.

  1. Prediction of physico-chemical properties does not add to the quality of the results and should be removed, unless authors can explain how these properties affect the antifungal activity of these Hsp90 inhibitors?

As introduced in the first part, the fungal wall serves as a mechanical barrier to numerous compounds. It was necessary to rule out the hypothesis that the lack of antifungal activity could be attributed to the incompatibility of the physicochemical properties of Hsp90 inhibitors with the fungal membrane-wall structure. Relying on the SWISS prediction tool used, the architecture of the fungal cell should not be a hindrance, while keeping in mind the potential involvement of transporters, as discussed in the previous section. Since the physicochemical properties are challenging to interpret, these are, indeed, only hypotheses.

  1. In addition to sequence alignment and conformation comparison, it would be nice to provide some molecular docking results to show how the sequence differences affect the Hsp90-inhibitor binding. It would be nicer if the molecular docking result could explain the activity differences of the tested inhibitors in Table 1.

That was not the objective of this work. However, more in-depth investigations are currently underway, and we hope to address this question in a future publication.

  1. The conclusion “the identification of selective CTD inhibitors of fungal Hsp90 could be a promising strategy for the development of innovative antifungal drugs” is based on “the greater sequence divergence in this domain”. However, the biological evaluation revealed that “the inhibitors binding to the C-terminal domain (CTD) did not show any antifungal activity, except one of them (ECGG, the multi-target one). ” Authors should indicate how to develop effective and species-selective antifungal Hsp90 inhibitors and provide a possible strategy in the manuscript.

We can assume that the identification of selective CTD inhibitors of fungal Hsp90 could provide very promising antifungal agents. One strategy would be to validate, first, species-selective antifungal activity through a phenotypic screening of the designed inhibitors. Afterwards, the selected compounds, endowed with suitable pharmacokinetic properties, will be engaged in a biochemical test against recombinant Hsp90 to validate, in a first step, the protein inhibition and then the CTD binding mode.

  1. There is no CONCLUSION part in this manuscript.

A concluding sentence has been added (line 480)

  1. There are many format errors in this manuscript.

Format errors related to margins, typeface, page numbering, and abbreviation reminders have been corrected.

We greatly appreciate your input, as it has significantly contributed to the enhancement of our manuscript. Your insights have been instrumental in improving the quality of our research, and we are grateful for your time and effort in reviewing our work. If you have any further questions or require additional information, please do not hesitate to reach out. Thank you for your support and guidance.

Sincerely

Reviewer 2 Report

Comments and Suggestions for Authors

1. 90-kDa Heat-Shock Protein (Hsp90) repeated in abstract and introduction second para. first time expansion followed by shortcut type.. At line 39. remove " 90-kDa Heat-Shock Protein" just Hsp90 enough.

2. Line 34     Spp? means

3. Page 4. N-terminal. N should be italic

4. did you used any reference durg to compare these results? If so. what are the merits of your results compare to referee

5. The major concern is references. seems not recent mostly. try to reduce and add last five years references as much as

Over all the research  is worth to publish 

Author Response

Dear reviewer,

We appreciate your thorough review and the valuable comments you provided. We greatly appreciate your input, as it has significantly contributed to the enhancement of our manuscript. Your insights have been instrumental in improving the quality of our research, and we are grateful for your time and effort in reviewing our work. If you have any further questions or require additional information, please do not hesitate to reach out. Thank you for your support and guidance.

To follow up on your comments:

  1. 90-kDa Heat-Shock Protein (Hsp90) repeated in abstract and introduction second para. first time expansion followed by shortcut type.. At line 39. remove " 90-kDa Heat-Shock Protein" just Hsp90 enough.

This error has been corrected throughout the text.

  1. Line 34     Spp? Means

Candida “sp" generally refers to the genus in a general sense. Candida “spp" implies that multiple species are involved and that we can specify which ones.

  1. Page 4. N-terminal. N should be italic

This error has been corrected throughout the text, and also concerning “C-terminal”

  1. did you used any reference durg to compare these results? If so. what are the merits of your results compare to referee

In order to address this comment, we conducted an additional experiment (IC50 of voriconazole, as presented in Table 1), explaining a in our response. We apologize for any inconvenience this may have caused. Thank you for your understanding. The IC50 values obtained with the Hsp90 inhibitors are not lower than those of voriconazole. However, we remain convinced that Hsp90 is an interesting target in yeast. Therefore, we subsequently formulated hypotheses to understand the reasons for this lack of effectiveness, with the goal of exploring avenues for improvement.

  1. The major concern is references. seems not recent mostly. try to reduce and add last five years references as much as

Some references have been removed as they were redundant (2, 3, 7, 18, 19, 20, 28) while retaining the most recent ones. However, we would like to keep references to several studies dating back over 5 years because they represent the gold standard articles in the field. In particular, all the studies from the Canadian team led by Leah E. Cowen, which have played a significant role in elucidating the structure of fungal Hsp90 and its involvement in fungal intracellular machinery.

Sincerely

Round 2

Reviewer 1 Report

Comments and Suggestions for Authors

Thank you for the revision. Previous comments have been well addressed. However, I still have some concerns about this manuscript. 

1. In Table 1, IC50 values for many isolates are 0.25+- 0.00, which means that VRC has the same inhibition potency against 15 isolates, and exactly the  same results were obtained in at least 3 independent experiments. Could you please possibly explain this coincidence?  

2. IC50 values are not consistant in Table 1 and Table 2.

3. A blank space is required between number and unit. For example, 100 µM.

Author Response

Dear Reviewer,

We would like to express our sincere appreciation for your thorough review and constructive feedback on our manuscript.

  1. In Table 1, IC50 values for many isolates are 0.25+- 0.00, which means that VRC has the same inhibition potency against 15 isolates, and exactly the same results were obtained in at least 3 independent experiments. Could you please possibly explain this coincidence? 

Indeed, it was an error during the transcription of the means. The IC50 values were <0.25 µM (the lowest concentration tested for voriconazole), which is why the standard deviations were zero. The IC50 values have been corrected in Table 1, and the standard deviations have been removed.

  1. IC50 values are not consistant in Table 1 and Table 2.

In Table 1, the values corresponded to the IC50 of the molecules incubated alone. In Table 2, the values corresponded to the IC50 of Hsp90 inhibitors combinations with the reference molecule, followed by the calculation of the resulting FICI. For example, for geldanamycin against the fluconazole resistant CAAL2:

Geldanamycine IC50 alone was 3.5 ± 1.2 µM (FICI was calculated with the lower value of 2.3 µM to be as restrictive as possible)

Fluconazole IC50 alone was > 100 µg/mL (FICI was calculated with the lower value of 101 µg/mL to be as restrictive as possible)

Geldanamycine IC50 in combination with fluconazole was 1 µM

Fluconazole IC50 in combination with geldanamycine was 0,9 µg/mL

FICI(geldanamycine) = 1 / 2,3 = 0,43

FICI(fluconazole) = 0,9 / 101 = 0,0089

FICI(combination) = FICI(geldanamycine) + FICI(fluconazole) 0,44

  1. A blank space is required between number and unit. For example, 100 µM.

Spaces have been introduced line 169, line 171 and line 250

Thank you for your time and effort in evaluating our work. Your expertise and thoughtful have significantly enhanced the clarity and quality of our manuscript. If you have any further comments or concerns, please do not hesitate to let us know.

We look forward to hearing your final evaluation and hope that our revised manuscript aligns more closely with the expectations of the journal.

Sincerely

Round 3

Reviewer 1 Report

Comments and Suggestions for Authors

The authous have addressed all of my concerns with the original manuscript. I recommend it to be published in Microorganisms.